# The Potential Use of Marine Microalgae and Cyanobacteria in Cosmetics and Thalassotherapy

**M. Lourdes Mourelle \* , Carmen P. Gómez and José L. Legido**

Applied Physics Department, University of Vigo, Campus Lagoas-Marcosende s/n, 36310 Vigo, Spain; carmengomez@uvigo.es (C.P.G.); xlegido@uvigo.es (J.L.L.)
* Correspondence: lmourelle@uvigo.es; Tel.: +34-696-413-531

**Abstract:** The use of microalgae and cyanobacteria for nutritional purposes dates back thousands of years; during the last few decades, microalgae culture has improved to become one of the modern biotechnologies. This has allowed high amounts of algal biomass to be obtained for use in different applications. Currently, the global production of microalgae and cyanobacteria is predominately aimed at applications with high added value given that algal biomass contains pigments, proteins, essential fatty acids, polysaccharides, vitamins, and minerals, all of which are of great interest in the preparation of natural products, both as food and in cosmetics. Hence, the bioactive components from microalgae can be incorporated in cosmetic and cosmeceutical formulations, and can help achieve benefits including the maintenance of skin structure and function. Thalassotherapy involves using seawater and all related marine elements, including macroalgae, however, there has been limited use of microalgae. Microalgae and cyanobacteria could be incorporated into health and wellness treatments applied in thalassotherapy centers due to their high concentration of biologically active substances that are of interest in skin care. This paper briefly reviews the current and potential cosmetic and cosmeceutical applications of marine microalgae and cyanobacteria compounds and also recommends its use in thalassotherapy well-being treatments.

**Keywords:** microalgae; cyanobacteria; cosmetics; cosmeceutics; thalassotherapy

## 1. Introduction

Ocean seawater is an environment where numerous organisms develop, some of which have been exploited for years to obtain new nutritional components and pharmacological molecules. However, marine biodiversity is so vast that it will take many more years for it to be fully explored. Over the last few decades, researchers of natural products have turned their attention to the marine environment as a rich source of plants, animals, and micro-organisms, which due to their adaptation to this unique environment, produce a large variety of primary and secondary metabolites with numerous and varied activities, for example, working against cancer and inflammation, or with anti-viral and immunomodulating actions [1].

The use of microalgae for nutritional purposes dates back thousands of years; from China, where *Nostoc* was used to survive famine, to Chad and Mexico where species of microalgae and cyanobacteria (*Arthrospira*, *Spirulina*) have been used by humans for thousands of years [2–5]. Despite these traditional uses, microalgae culture is still a modern biotechnology [5].

Various articles have reviewed the main active ingredients derived from cyanobacteria and microalgae. Cyanobacteria (formerly known as blue-green algae) are found in a variety of habitats as they can grow in freshwater and saltwater, as well as in marine environments. Although prokaryotic organisms, they are also able to carry out photosynthesis [6], and are considered as one of the earliest photosynthesizers on the planet. The most studied are *Nostoc*, *Spirulina (Arthrospira)*,

and *Aphanizomenon*, with the most commonly harvested components being carotenoids, chlorophyll, phycocyanins, amino acids, minerals, among many others [2].

Microalgae are prokaryotic and eukaryotic photosynthetic micro-organisms. There are two groups of prokaryotes (Cyanophyta and Prochlorophyta) and different divisions of eukaryotes (Chlorophyta, Rhodophyta, Phaeophyta, Bacillariophyta, and Chrysophyta) [7]. They are generalist and can therefore develop in freshwater, marine and highly saline environments.

Currently, the global production of microalgae and cyanobacteria is primarily aimed at applications with high added value [5,8], given that algal biomass contains pigments, proteins, essential fatty acids, polysaccharides, vitamins, and minerals which are of great interest in the preparation of natural products, both as food and in cosmetics.

## 2. Use of Microalgae and Cyanobacteria for Health Applications

Although the principal research into bioactive components from the marine environment and their health applications has stemmed from studying macroalgae, recently, this research has extended into microalgae as it is easier to cultivate.

Interest in harvesting bioactive components from microalgae and cyanobacteria to be used in health applications has increased in line with research efforts in the quest for molecules to treat various ailments such as diabetes, high blood pressure, and other disorders related to metabolism and the immune system.

One species of microalgae that has been studied for use in food, or for harvesting active ingredients for pharmacological use, is the freshwater *Haematococcus pluvialis*, which is cultivated and exploited industrially to obtain astaxanthin, considered a 'super antioxidant' [9]. Another widely cultivated microalgae is *Dunaliella tertiolecta*, which is highly tolerant to saline environments and is used to harvest ß-carotenes, although it is also known for producing the pigment violaxanthin, which has anticarcinogenic properties [10].

Species of the genus *Porphyridium* are used to obtain phycoerythrobilins and phycocyanobilins, protein colorings that are considered to be alternatives to synthetic colorings due to their marginal toxicity [11], and also for harvesting polysaccharides, which have antiviral properties [12]. Likewise, some species from the *Chlorella* genus (mainly *Chlorella vulgaris*) have been used to obtain chlorophyll for use as a pigment, but has also been attributed with anticarcinogenic properties [13]. Nevertheless, the most important substance in *Chlorella* is ß-1,3-glucan, which is an active immunostimulator as it rounds up free radicals and reduces lipids in the blood [3].

The species *Chlorella pyrenoidosa* and *Chlorella elipsoide* contain complex polysaccharides which have immunostimulating properties [14]. Other studies on *Chlorella* prove its antioxidizing, anti-inflammatory and analgesic abilities [15] due to the polysaccharides [16].

Harvesting polysaccharides from microalgae and cyanobacteria, in particular sulfated polysaccharides, is attracting growing interest due to the fact that many species create them under flexible cultivation conditions. Polysaccharides have demonstrated anti-inflammatory, immunomodulating, and antiviral properties as well as being used as joint lubricants. The most popular species and genera include *Tetraselmis* sp., *Isochrysis* sp., *Porphyridium cruentum*, and *Porphyridium purpureum*, in addition to the above-mentioned species of *Chlorella*, among others [17].

Another highly interesting aspect is the production of lipids such as triacylglycerol (TAG), which is of special interest for human consumption [18] as it has a composition of fatty acids similar to many vegetable oils, including some of high value such as eicosapentaenoic acid (EPA) and docosahexaenoic acid (DHA). Other species of microalgae that produce TAGs are *Nannochloropsis gaditana*, *Scenedesmus obliquus*, and the diatom *Thalassiosira pseudonana* [19].

Studies carried out on a marine diatom, *Odontella aurita*, which is rich in EPA and other bioactive molecules such as pigments, has shown that it has a beneficial effect on reducing metabolic syndrome risk factors (hyperlipidemia, platelet aggregation, and oxidative stress), attributing the effects to the synergy between the active components of this species [20].

Many other microalgae are producers of polyunsaturated fatty acids (PUFAs) such as *Phaeodactylum tricornutum*, *Monodus subterraneus*, *P. cruentum*, *Chaetoceros calcitrans*, *Nannochloropsis*, *Crypthecodinium cohnii*, *Isochrysis galbana*, and *Pavlova salina*. Due to this richness in unsaturated fatty acids, it has been suggested that microalgae can be utilized as an alternative and sustainable source of PUFAs for use in nutraceuticals [21]. Microalgae and cyanobacteria are increasingly used in food; the entire species is marketed as a food supplement in tablet, capsule, or powder form, although only a few species are used considering the thousands that are known about [22]. The following species can be found in the European Commission's database of new foods (Novel Food Catalogue) (http://ec.europa.eu/food/safety/novel_food/catalogue/search/public/index.cfm): *Arthrospira platensis*, *Aphanizomenon flos-aquae* var. *flos-aquae*, *Chlorella vulgaris*, *Chlorella pyrenoidosa*, and *Chlorella luteoviridis*.

However, the bioactive components of many other species can be found in foodstuffs as additives (mainly colorings), or as natural ingredients such as lipids (mainly PUFAs), sterols, proteins, amino acids, polysaccharides, polyphenols, and vitamins, and as food additives (colorings, thickeners, etc.).

This range of biologically active molecules has led both freshwater and marine microalgae and cyanobacteria to leave their mark in nutritional supplements and nutraceuticals formulations, although, as indicated, only a handful of species are recognized as ingredients for use in the food industry, these being the proteins, peptides and amino acids from *Chlorella vulgaris* and *Spirulina* [1,23], the lipids and fatty acids of *Haematococcus* and *Spirulina* [24], the ß-carotenes from *Dunaliella salina*, and the phycocyanobilins from *Spirulina* as anti-oxidants, to name a few [25].

Another nutritional aspect of interest is the potential use of the polysaccharides as a source of fiber and also as a prebiotic [17], though these are yet to be developed.

## 3. Uses of Microalgae and Cyanobacteria for Cosmetic Applications

Bearing in mind the diverse pharmacological activities attributed to different species of cyanobacteria and microalgae, it appears that many of the molecules responsible for these activities also act upon the skin, given that these are organisms whose physical integrity and mechanical properties are linked to its ability to regenerate and protect itself against external environmental conditions [26]. However, there have only been a few studies on this aspect, as summarized in Table 1.

**Table 1.** Bioactive compounds from microalgae and cyanobacteria and their potential uses in cosmetics.

| Bioactive Compounds | Microalgae/Cyanobacteria | Potential Activities and Uses in Cosmetics | References |
|---|---|---|---|
| Polysaccharides | *Chlorella* | Moisturizing and thickener agent | Jain et al., 2005 |
| Methanolic extracts of exopolysaccharides | *Arthrospira platensis* | Antioxidant | Raposo et al. 2015 |
| Chrysolaminarin | *Odontella aurita* | Antioxidant | Xia et al., 2014 |
| Sulfated polysacharides | *Porphyridium and Rhodella reticulata* | Antioxidant | Raposo et al., 2015 |
| ß-1,3-Glucan | *Chlorella Skeletonema Porphyridium Nostoc flegelliforme* | Free-radical collector Immune system booster Anti-inflammatory | Spolaore et al., 2006 Koller et al., 2014 Bin et al., 2013 Hamed, 2016 |
| ß-carotenes | *Dunaliella salina* | Antioxidant | Hamed, 2016 |
| Asthaxanthin | *Haematococcus pluvialis* | Antioxidant Sunscreen protection | Hamed, 2016 Koller et al., 2014 |
| Phycocyanobilin phycoerythrobilin | *Spirulina Porphyridium* | Antioxidant Pigment for eye-liner and lipsticks | Hamed, 2016 |

**Table 1.** *Cont.*

| Bioactive Compounds | Microalgae/Cyanobacteria | Potential Activities and Uses in Cosmetics | References |
|---|---|---|---|
| β-Cryptoxanthin | *Dunaliella salina* | Anti-inflammatory Promote Hialuronan synthesis | Tang and Suter, 2011 |
| Chlorophyll | *Chlorella* sp. | To mask odors in dentifrices and deodorants | Hosikian et al., 2010 |
| Canthaxanthin | *Nannochloropsis salina* *Nannochloropsis oculata* *Nannochloropsis gaditana* | Tanning cosmetics and cosmeceutics | Koller et al., 2014 |
| Phycocyanin | *Porphyridium cruentum* *Spirulina platensis* | Eye-shadows | Bermejo et al., 2003 Arad and Yaron, 1992 |
| Lycopene | *Anabaena vaginicola* | Antioxidant Anti-ageing Sunscreen | Singh et al., 2012 Hashtroudi et al., 2013 |
| Scytonemin | Marine cyanobacteria | Sunscreen | Takamatsu et al., 2003 |
| Vitamin E (α-Tocopherol) | *Dunaliella tertiolecta* *Tetraselmis suecica* | Antioxidant | Hashtroudi et al., 1999 |
| Biopterin glucose | Marine planktonic cyanobacterium | Sunscreen | Matsunaga et al., 1993 |
| Ectoine | *Halomonas elongata* *Halomonas boliviensis* *Brevibacterium epidermis* *Chromohalobacter israelensis* *Chromohalobacter salexigens* | Immune protection UV protection Stress protection Moisturizing agent | Kim et al., 2008 Shivanand and Mugeraya, 2011 |
| Phytohormones (auxin, abscisic acid, cytokinin, ethylene, gibberellins) | Broad spectrum of microalgal lineages *Nannochloropsis oceanica* | Anti-ageing | Lu and Xu, 2015 Michelet et al., 2012 |
| Micosporine-like amino acids | Cyanobacteria | Sunscreen | Llewellyn and Airs, 2010 Singh, 2017 |
| *Chlorella vulgaris* extracts | *Chlorella vulgaris* | Collagen repair (anti-ageing) | Koller et al., 2014 |
| Microalgae extracts | *Phaeodactylum tricornutum* *Scenedesmus vacuolatus* and *Chlorella kessleri* | Antioxidant Antioxidant | Morelli et al., 2004 Sabatini et al., 2009 |

Microalgae and cyanobacteria generate components that may be of interest to the cosmetic industry (personal care products) [27]. The active ingredients extracted from certain microalgae are currently used in cosmetic and cosmeceutical products, although more research into their action mechanisms are needed.

Cosmetics are products aimed at improving the structure, morphology, and appearance of skin, with the assistance of excipients and active ingredients adapted to different skin types (normal, oily, combination, sensitive, etc.). Cosmeceutics, although a term not officially recognized, are defined as "cosmetic products with biologically active ingredients purporting to have medical or drug-like benefits" [26].

The significance of active ingredients derived from microalgae is that they can be used to prevent blemishes, repair damaged skin, help seborrhea, and inhibit the inflammation process. In addition, extracts from microalgae have various bioactive substances which accelerate the healing process and maintain skin moisture [28].

Among the active ingredients extracted from microalgae that have potential uses in cosmetics are polysaccharides which, just as those from macroalgae, are used as gelling agents and thickeners in different cosmetic formulas, as well as for moisturizing (for example, the genus *Chlorella*) [29].

Equally, the fact that some of these polysaccharides, especially ß-1,3-glucan, are good free-radical collectors and are active immunostimulators, makes them good candidates for use in skincare cosmetics, especially in preventing external ageing (linked to environmental factors and oxidation caused by free radicals) [3,30]. Furthermore, given that ß-1,3-glucan from plants and fungi appears to act as an anti-inflammatory [14], presumably this action would also be present in microalgal glucans, which would constitute an interesting field of research for the cosmetic sector targeting sensitive and reactive skin types. Species rich in ß-glucans are from the *Chlorella* genus and the *Skeletonema* diatom, as well as *Porphyridium* and *Nostoc flegelliforme* [23]. Skin is equipped with a very complex antioxidant system that protects it from oxidative damage due to intrinsic and extrinsic factors. However, the natural antioxidant pool can be compromised or overwhelmed. Topical antioxidants have been demonstrated to protect the skin from oxidative stress and damaging free radicals produced intrinsically by normal cellular metabolism or through exposure to UV light.

The antioxidizing potential of colorings produced by microalgae and cyanobacteria may also be of huge value to the cosmetic industry, as they can be used as cosmetic antioxidants and also as natural colorants. This mainly includes the photosynthetic pigments such as carotenoids, for example, the ß-carotenes from *Dunaliella salina*; astaxanthin from *Haematococcus pluvialis* (red color), has an antioxidant that is ten times more powerful than other carotenoids (such as ß-carotenes, zeaxanthin, etc.), and 100 times stronger than alpha-tocopherol [23]; phycocyanobilins (blue pigment); and phycoerythrobilins from *Spirulina* and *Porphyridium*, which contain antioxidants and could be used in the decorative cosmetics industry (eye-liner and lipstick) [23]. Phycobiliproteins obtained from *Porphyridium aerogineum* are also used as colorants in food and cosmetics; this pigment does not change with pH (4 to 5) and the color remained constant under light [31].

Astaxanthin is considered the most powerful natural antioxidant, hence a highly efficient scavenger of free radicals. In human metabolism, astaxanthin is also of importance for skin protection against UV-induced photo-oxidation [30] so it can be used in natural sunscreen cosmetics. *Phaeodactylum tricornutum*, a diatom that lives in marine environments, has also been proven as an antioxidant [32], as well as the microalgae *Scenedesmus vacuolatus* and *Chlorella kessleri* [33].

A review of the activity of carotenoids from microalgae and cyanobacteria shows extensive variability in the bioactive components at play including antioxidants, photoprotection, anti-inflammatory, and anti-allergenic properties, with results for skin as well [34]. It has been observed that ß-cryptoxanthin (a carotene found in *Dunaliella salina*, among other microalgae) [35], as well as being an anti-inflammatory, is also able to induce the synthesis of hyaluronic acid [36], a glycosaminoglycan involved in skin hydration.

Other pigments from microalgae such as chlorophyll can easily be extracted [13] and used in cosmetics, for example, in deodorants, due to their ability to mask odors, as well as in toothpastes and hygiene products.

Another pigment, canthaxanthin, is commercialized in tanning pills [30]; this pigment is mainly obtained from *Nannochloropsis* sp. like *Nannochloropsis salina*, *Nannochloropsis oculata*, or *Nannochloropsis gaditana* species [37]. There is also interest in the use of some pigments in make-up formulations; phycocyanins, produced by thermophilic blue-green algae, can be used for the formulation of eye shadows [38], and the pink and purple colors included in cosmetics could also be formulated from the natural colorants extracted from red microalgae [39].

Other bioactive compounds are less known such as Biopterin glucose (an UV-A absorbing chromophore), which is a pigment produced from a marine planktonic cyanobacteria, and protects the skin from the adverse effects of the UV-A radiation, hence its use in the formulation of sunscreen cosmetics [40]. Additionally, scytonemin is a carotenoid produced by marine cyanobacteria that can be also used as an UV sunscreen cosmetic material as it has demonstrated antioxidant activity in different assays [41].

Lycopene belongs to the family of carotenoids and is an efficient antioxidant that can neutralize oxygen derived free radicals. It is considered as the most potent natural antioxidant, more so than tocopherol, β-carotene and lutein, as well as a sunburn preventing agent, and can be used as a sunscreen agent [42]. Hashtroudi et al. [43] found that the cyanobacteria *Anabaena vaginicola* had a substantially higher content of lycopene than all previously reported natural sources. Today, lycopene is used in personal care formulations as an anti-aging agent, so cyanobacteria and microalgae could be a source of lycopene for cosmetic uses.

Microalgae could provide sources of vitamin E. *Dunaliella tertiolecta* and *Tetraselmis suecica*, which are widely used in aquaculture as feed for fish and mollusk larvae, produce relatively high concentrations of α-tocopherol and vitamins [44]. Vitamin E is also considered as an effective antioxidant and is widely used in cosmetic formulations.

Related to skin ageing supplements and cosmeceutics, extracts from *Chlorella vulgaris* have been reported to support collagen repair mechanisms [31], and Raposo et al. [45] reported that sulfated PS (sPS) produced and secreted by marine microalgae had shown the capacity to prevent the accumulation and activity of free radicals and reactive chemical species. Therefore, sPS might act as protecting systems against these oxidative and radical stress agents. The sPS from *Porphyridium* and *Rhodella reticulate* exhibited antioxidant activity that was dose-dependent. Methanolic extracts of exopolysaccharides (EPS) from *A. platensis* also exhibit a very high antioxidant capacity [45].

Furthermore, the microalgae *Odontella aurita*, from which the glucan, chrysolaminarin (demonstrating antioxidant activity) is extracted, should also be mentioned [46].

Ectoine (1,4,5,6-tetrahydro-2-methyl-4-pyrimidinecarbo-xylic acid) is one of the most common osmotic solutes in the bacteria domain. Ectoine provides multiple cosmetic benefits such as immune protection, cell protection, UV protection, and membrane protection [28], as well as being a stress protective agent. It is claimed that it counteracts the effects of UV-A induced and accelerated skin ageing, and is therefore being used as a dermatological cosmetic additive in moisturizers for the care of aged, dry, or irritated skin. Ectoines can be found in halophilic bacteria such as *Halomonas elongata*, *Halomonas boliviensis*, *Brevibacterium epidermis*, *Chromohalobacter israelensis*, or *Chromohalobacter salexigens* [47]. Hence, halophilic bacteria could also be a source of components for skin care products.

Another field of interest is the production of phytohormones, as it has been demonstrated that different genus of cyanobacteria and algae accumulate and release a diverse group of phytohormones that are involved in plant growth and development [48]. Phytohormones, including auxin, abscisic acid, cytokinin, ethylene (ET), and gibberellins, have been found in a broad spectrum of microalgal lineages. Although the functional role of microalgal endogenous phytohormones remains elusive, molecular evidence from the oleaginous microalgae *Nannochloropsis oceanica* suggests that endogenous abscisic acid and cytokinin are functional, and that their physiological effects are similar to those in higher plants [49]. That means the microalgal phytohormones could play a role in counteracting signs of skin ageing [50], a promising future in skin care cosmetics.

It is also worth mentioning the research in obtaining micosporine-like amino acids from cyanobacteria [51], which are similar to those obtained from macroalgae, for their potential use in sunscreens; and other photo-protective compounds such as scytonemin (a dimer of indolic and phenolic subunits), which is capable of reducing the risk of damage caused by UV light [48].

As some microalgae extracts can be used in skin care products (e.g., anti-aging creams, emollients, protecting, refreshing, or regenerating skin care products, antioxidant and anti-irritant products, and deodorant products), cosmetic companies have even invested in their own microalgal production

system (e.g., Louis Vuitton Moët Hennessy, Paris, France, and Danial Jouvance, Carnac, France); and other commercialized products including a liposome-based product containing a photolyase from blue-green algae, *Anacystis nidulans*, manufactured by the American company (AGI Dermatics). The extract from *Chlorella vulgaris* stimulates collagen synthesis in the skin and can be used in products supporting tissue regeneration and wrinkle reduction (Dermochlorella, Codif, St. Malo, France) [3].

Other examples include products from extracts of the marine microalgae *Nannochloropsis oculata*, which have been launched by Pentapharm (Basel, Switzerland), for their excellent skin-tightening effects. Additionally, Blue Retinol[TM], an alga extract from *D. salina*, stimulates skin cell growth and proliferation [27]. Another product, SILIDINE® by Greentech USA, was developed from the purple-red alga *Porphyridium cruentum* using the technique of metabolic induction with the claim that it improves vascular tonicity, helps decrease heavy leg syndrome, improves the skin aspect, and helps decrease rosacea effects and redness. Furthermore, this product is recommended for uniformity and radiance of complexion, vascular tonicity enhancement, body tonicity, sensitive skin and against redness. Additionally, a protein-rich extract from *Arthrospira* claims to repair the signs of skin aging, exerts a tightening effect, and prevents stria formation (Protulines, Exsymol S.A.M., Monaco) [52].

Recently, new patents are being registered as new compositions suitable for topical administration comprising microalgal exopolysaccharide particles, e.g., the exopolysaccharides produced by microalgae of the genus *Parachlorella*, for improving the health and appearance of skin by Solazyme Inc. (San Francisco, CA, USA), and others comprising of biomass from *Chlorella protothecoides* (Solazyme Inc.), or microalgae biomass combined with microalgal oil by Terravia Holdings, Inc. (San Francisco, CA, USA). The latter company has recently launched GoldenChlorella™ and AlgaPür™ Algae Oils that claim to deliver strong cosmetic benefits to skin and hair.

On the other hand, Givaudan (Switzerland) launched the *Phaeodactylum tricornutum* extract, an anti-aging agent. It is a lipidic extract of a mono-cellular micro-alga belonging to the diatomophyceae order, *P. tricornutum* that claims to stimulate cell detoxification from oxidized proteins through a specific enzymatic system, the proteasome, thus preventing chronological and premature aging by limiting the accumulation of harmful proteins.

In summary, different bioactive compounds could be considered as potential candidates as active ingredients in cosmetics and cosmeceutics for skin care. Figure 1 illustrates the various cosmetic and cosmeceutical applications of marine microalgae and cyanobacteria-derived bioactive compounds.

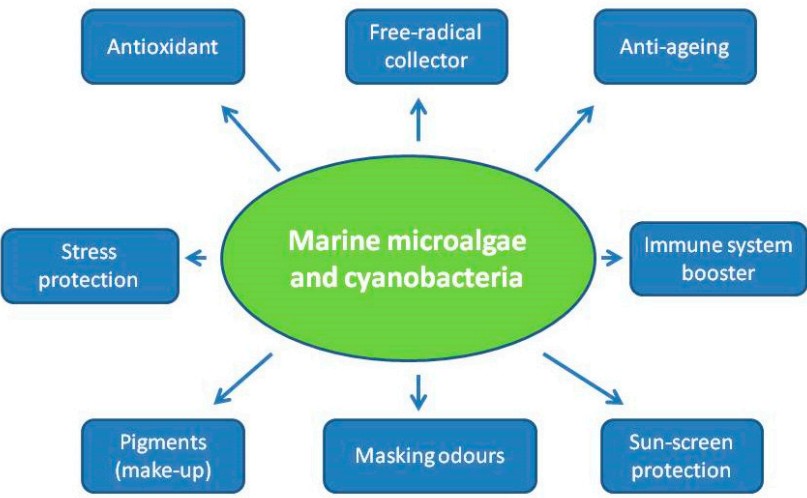

**Figure 1.** Potential benefits of marine microalgae and cyanobacteria-derived active ingredients in skin care.

By way of demonstrating the interest by the cosmetic industry, more than 120 examples of cosmetic substances or ingredients can be found in the Cosmetic Ingredient Database (CosIng) https://ec.europa.eu/growth/sectors/cosmetics/cosing_es. Some examples are shown in Table 2.

**Table 2.** INCI names (International Nomenclature of Cosmetic Ingredients) and CAS number (Chemical Abstracts Service) of cosmetic substances and ingredients on CosIng (Cosmetic Ingredient Database).

| INCI Name | Description |
|---|---|
| ALGAE EXOPOLYSACCHARIDES | Algae Exopolysaccharides are exopolysaccharides released by the fermentation of various species of microalgae of the divisions, *Rhodophyta* and *Chlorophyta*. |
| PARACHLORELLA BEIJERINCKII EXOPOLYSACCHARIDES | Parachlorella Beijerinckii Exopolysaccharides are exopolysaccharides produced through the fermentation of the microalgae *Parachlorella beijerinckii*. |
| APHANOTHECE SACRUM POWDER | Anacardoyl Tripeptide-35 is the product obtained by the reaction of anacardic acid and Tripeptide-35, Cyanobacteriaceae. |
| ARTHROSPIRA EXTRACT | Arthrospira Extract is an extract of the cyanobacterium, *Arthrospira maxima*. |
| ARTHROSPIRA PLATENSIS CULTURE CONDITIONED MEDIA | Arthrospira Platensis Culture Conditioned Media is the growth media removed from cultures of *Arthrospira platensis* after several days of growth, Phormidiaceae. |
| CYANOBACTERIUM APONINUM FERMENT | Cyanobacterium Aponinum Ferment is the product obtained by the fermentation of *Cyanobacterium aponinum*. |
| ODONTELLA AURITA EXTRACT | Odontella Aurita Extract is an extract of the phytoplankton *Odontella aurita*, Bacilaiophyceae. |
| ODONTELLA AURITA OIL | Odontella Aurita Oil is the oil obtained from the phytoplankton *Odontella aurita*, Bacilaiophyceae. |
| PARALLELOSTROMBIDIUM SICULUM EXTRACT | Parallelostrombidium Siculum Extract is the extract of the plankton, *Parallelostrombidium siculum*, Strombidiidae. |
| PLANKTON EXTRACT (CAS No. 91079-57-1) | Plankton Extract is an extract obtained from marine plankton. |
| TETRASELMIS SUECICA EXTRACT | Tetraselmis Suecica Extract is an extract of the plankton *Tetraselmis suecica*. |
| CHLORELLA ELLIPSOIDEA EXTRACT | Chlorella Ellipsoidea Extract is the extract of the alga, *Chlorella ellipsoidea*, Chlorellaceae. |
| CHLORELLA EMERSONII EXTRACT (CAS No. 223749-78-8) | Chlorella Emersonii Extract is an extract of the Alga, *Chlorella emersonii*, Oocystaceae. |
| CHLORELLA VULGARIS CALLUS CULTURE EXTRACT | Chlorella Vulgaris Callus Culture Extract is the extract of a culture of the callus of the alga, *Chlorella vulgaris*, Chlorellaceae. |
| PHORMIDIUM UNCINATUM EXTRACT | Phormidium Uncinatum Extract is the extract of the Alga, *Phormidium uncinatum*. |
| PHORMIDIUM FERMENT | Phormidium Ferment is the product obtained through fermentation by the microorganism, *Phormidium*. |
| APHANIZOMENON FLOS-AQUAE EXTRACT | Aphanizomenon Flos-Aquae Extract is the extract of the alga, *Aphanizomenon flos-aquae*, Nostocaceae. |
| ALPHA-GLUCAN OLIGOSACCHARIDE | Alpha-Glucan Oligosaccharide is a glucose oligomer exhibiting a degree of polymerization ranging from 2–10. It is prepared by the action of a *Leuconostoc mesenteroides* glucosyl transferase on sucrose. |
| NOSTOC FLAGELLIFORME EXTRACT | Nostoc Flagelliforme extract is an extract of the Alga, *Nostoc flagelliforme*, Nostocaceae. |
| DUNALIELLA BARDAWIL EXTRACT | Dunaliella Bardawil Extract is an extract of the Alga, *Dunaliella bardawil*, Dunaliellaceae. |

**Table 2.** *Cont.*

| INCI Name | Description |
| --- | --- |
| DUNALIELLA SALINA/HAEMATOCOCCUS PLUVIALIS EXTRACT | Dunaliella Salina/Haematococcus Pluvialis Extract is the extract of the alga, *Dunaliella salina* and *Haematococcus pluvialis*. |
| PORPHYRIDIUM CRUENTUM EXTRACT (CAS No. 223751-77-7) | Porphyridium Cruentum Extract is an extract of the Alga, *Porphyridium cruentum*, Porphyridaceae. |
| THALASSIOSIRA PSEUDONANA EXTRACT | Thalassiosira Pseudonana Extract is the extract of the alga, *Thalassiosira pseudonana*. |
| NANNOCHLOROPSIS OCULATA POWDER | Nannochloropsis Oculata Powder is the powder obtained from the dried, ground alga, *Nannochloropsis oculata*. |
| SKELETONEMA COSTATUM EXTRACT | Skeletonema Costatum Extract is an extract of *Skeletonema costatum*. |
| DUNALIELLA SALINA/ISOCHRYSIS GALBANA/NANNOCHLOROPSIS GADITANA/PHAEODACTYLUM TRICORNUTUM/TETRASELMIS CHUII EXTRACT | Dunaliella Salina/Isochrysis Galbana/Nannochloropsis Gaditana/Phaeodactylum Tricornutum/Tetraselmis Chuii Extract is the extract of the whole plants of *Dunaliella salina*, *Isochrysis galbana*, *Nannochloropsis gaditana*, *Phaeodactylum tricornutum* and *Tetraselmis chuii*. |
| PSEUDANABAENACEAE FERMENT FILTRATE | Pseudanabaenaceae Ferment Filtrate is a filtrate of the product obtained by the fermentation of a growth media by the microorganism, Pseudanabaenaceae. |
| RHODELLA VIOLACEA EXTRACT | Rhodella Violacea Extract is the extract of the alga, *Rhodella violacea*, Rhodellaceae. |
| HALOMONAS ANTICARIENSIS FERMENT EXTRACT FILTRATE | Halomonas Anticariensis Ferment Extract Filtrate is a filtrate of an extract of the product obtained by the fermentation of the bacteria, *Halomonas anticariensis*. |

## 4. The Application of Marine Microalgae in Thalassotherapy

Microalgae are of great interest in the field of thalassotherapy due to their aforementioned properties and wealth of unsaturated fatty acids (EPA and DHA), vitamins A, B$_1$, B$_2$, B$_6$, B$_{12}$, C, E, nicotinamide, biotin, folic acid, and pantothenic acid as they can be used as food (for special diets), nutraceuticals (as nutritional supplements) or cosmetics (as emollients and protective agents).

Thalassotherapy involves using seawater and all related marine elements (marine environment, algae and plankton, sea mud and sand) for therapeutic and preventive purposes. Thalassotherapy centers are increasingly including anti-stress and revitalizing programs which consist of hydrothermal techniques combined with marine cosmetics, massage techniques, and various additional activities (sessions on nutrition, mindfulness, relaxation, etc.) in their lists of wellbeing treatments to aid recovery from the negative effects of modern living such as stress, anxiety, and tiredness [53].

Peloids are blends of an inorganic or organic solid base and thermal water, which after a maturation process, are used for therapeutic purposes [54]. Those made of seawater are also called marine mud, or sea mud. For wellbeing purposes, peloids are mixed with macroalgae or any other natural product (e.g., herbs) and applied on the body in the form of poultices or cataplasms.

Although macroalgae have been used for decades in thalassotherapy centers, this has not been the case for microalgae. At the time of writing this review, publications related to the uses of microalgae in thalasso centers were not available. The only research group currently working on this topic is the FA2 group of the Applied Physics department at the University of Vigo, Spain, which has presented its results at different conferences. The FA2 group published a pilot experiment to cultivate marine microalgae for application in thalassotherapy [55] that entailed cultivating microalgae in seawater in the Talaso Atlántico Thalassotherapy Centre (in Oia, Pontevedra, Spain). This subsequently led to the

development of two products for application in wellbeing programs: a microalgae bath and a peloid (marine mud), which were also developed from microalgae.

To cultivate the microalgae the marine specie *Nannochloropsis gaditana* was selected and cultivated in a bioreactor with temperature, pH, $CO_2$, and air control. The conditions were set as follows: temperature between 20–23 °C, and a pH between 6.5 and 7.5. A picture of the cultivation system is shown in Figure 2.

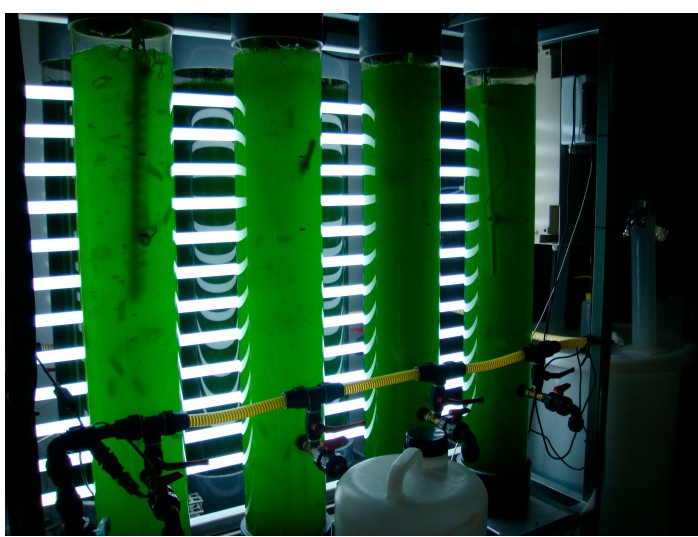

**Figure 2.** Photobioreactor with microalgae (Applied Physics department, University of Vigo).

The microalgae used had a high level of proteins (40–50% of its dry weight) and a high content of unsaturated fatty acids (PUFAs) of which between 25% and 30% were EPA (20:5n-3) and 6–9% ARA (20:4n-6).

For use in thalassotherapy, the biomass of the microalgae was taken every month (obtaining around 10 kg of microalgae 25% concentration each month) and frozen for subsequent use.

The spa immersion was performed by adding 150 g of frozen microalgae to a whirlpool bath, setting it to 20 min of jets with full-body cover. The acceptability of this technique by users is still pending evaluation and the results are yet to be published.

To prepare the marine mud, commercial bentonite was used (supplied by Bentonitas Especiales S.A. and previously discussed in Casás et al. 2011, and Casás et al. 2013 [56,57]) and mixed with seawater. This mixture was compared with another of similar characteristics, but with triple-distilled water as the liquid substrate. The triple-distilled water was supplied by C.A.C.T.I. (Center for Scientific and Technological Research Support) and obtained using the MilliQ system (Millipore). The seawater came from Quinton Laboratories; a hypertonic water with dry residue of 35.614 mg/L (to 453.15 K for 24 h), with a pH of 7.9 measured at 298.15 K [57] (Figure 3).

In addition to the physical–chemical breakdown, a study of the skin biometrology of 20 volunteers was carried out on an application of the microalgal peloid to their forearms, with measurements taken before and after daily applications of 15 min a day over two weeks. To measure the moisturizing property, a Corneometer CM 825 by Courage-Khazaka was used, and to gauge elasticity and fatigue, an MPA 580 multisensor by Courage-Khazaka was employed [58]. The results from this study showed that the microalgal peloid improved skin moisturize and elasticity, and above all, fatigue; therefore, the authors considered that its use in cosmetics and thalassotherapy treatments may be of interest.

Despite the lack of research in the use of microalgae in the field of thalassotherapy, these experiences allow the visualization of a commercial future in the cultivation and subsequent cosmetic applications of microalgae in the thalassotherapy center itself.

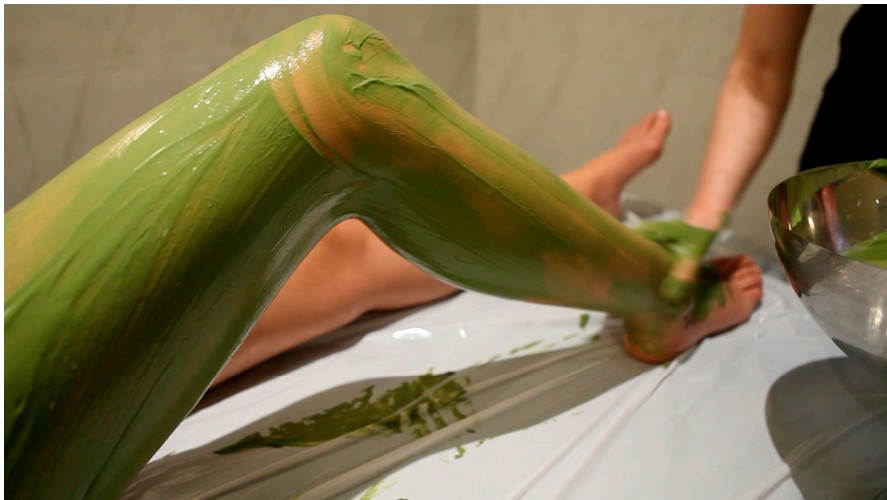

**Figure 3.** Marine mud (microalgal peloid) application (Talaso Atlántico, Oia, Pontevedra, Spain).

## 5. Conclusions and Perspectives

Microalgae and cyanobacteria are an exceptional source for a variety of bioactive compounds. Among them, polysaccharides, carotenes, lipids, and proteins are recognized to have numerous positive effects on health.

The effects of microalgae and their nutraceutical derived products have been tested in different nutritional studies worldwide; however, there are very few studies regarding their cosmetic applications. The more studied species are *Chlorella*, *Spirulina*, *Nannochloropsis*, *Porphyridium*, *Nostoc*, and *Dunaliella* and their potential for preventing skin ageing, protecting against UV light damage, and oxidative stress should increase interest and promote research activities into their value as cosmetic and cosmeceutical ingredients.

Another field of interest could be the production of natural colorants for beauty cosmetics (e.g., lipsticks, eye shadows, eye liners, etc.) as some microalgae and cyanobacteria are rich in pigments such as phycocyanins, phycocyanobilins, and phycoerythrobilins. Thus, more intensive research on identifying new compounds from microalgae and cyanobacteria could be of interest to the cosmetic industry.

Microalgae are specifically cultivated at an industrial scale for the production of high quantities of different compounds or specific bioactive molecules (e.g., EPA and DHA) [59]. Hence, the commercial exploitation of microalgae and cyanobacteria for cosmetic uses could also be improved by modifying biomass production to optimize the manufacture of specific cosmetic ingredients.

In this review, we have shown that there is a potential market for the cosmetic and cosmeceutical uses of the bioactive compounds of microalgae and cyanobacteria, with a specific market for use in thalassotherapy. Thus, research in this field should be promoted.

In conclusion, with the increase in demand for natural products for skin care and wellbeing treatments in spa and thalassotherapy centers, microalgae may be a significant source of substances with beneficial effects for skin health. Its wealth of polysaccharides, carotenoids, unsaturated fatty acids, vitamins, and other antioxidants that protect against UV radiation, etc., makes them potential raw materials for the development of cosmetics and cosmeceutics, but also for thalasso products; therefore, research into these ingredients and their uses must be promoted and extended.

**Author Contributions:** The individual contribution and responsibilities of the authors were as follows: M. Lourdes Mourelle and Carmen P. Gomez conceived and designed the paper, and performed the search of literature; all three authors analyzed the literature data; and M. Lourdes Mourelle wrote the paper.

**Conflicts of Interest:** The authors declare no conflict of interest.

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
