# Peer review of "The Potential Use of Marine Microalgae and Cyanobacteria in Cosmetics and Thalassotherapy"

_cosmetics, doi:10.3390/cosmetics4040046_

Round 1

Reviewer 1 Report

This work reviews the cosmetic and cosmeceutic interest of microalgae and cyanobacteria advantage. Furthermore, it includes a section in which some applications of marine microalgae and cyanobacteria in thalassotherapy are discussed. A list of relatively recent 51 publications were reviewed for the elaboration of this paper. Works like this are important since there is a lack of information in this area of research. Nevertheless, an exhaustive major revision should be done in order to be approved.

In general, English expression and punctuation should be reviewed throughout the text. Some phrases are very long and meaning is not clear. Repetition of words and use of capital letters should be edited. Paragraphs dealing with the same idea should be pooled. Some passages lack of academic writing style and are very casual, especially section 3 and 4. Names of microalgae and cyanobacteria have to be written in italics, first letter in capitals, the rest in lower case; ie. Odontella aurita. It is always recommended to hire an academic reviewer/editor.

Paragraphs need to be organized properly, considering to use subsections or subtitles. Some ideas are: separate microalgae and cyanobateria; organize sections by activity or bioactive compounds, split information about thalassotherapy and cultivation. 

There is some information about nutrition in the section of cosmetics, is it possible to organize this properly maybe in a specific section for this? Please organize.

There are physicochemical and technological properties, and biological activities. Please, organize.

Line 9  The abstract has to be rewritten and include all the review sections. Something about cosmeceutics and thalassotherapy should be included. 

Line 20 The introduction lacks of information of these two keywords, too.

Consider to change the suffix 'beta' by the letter β along the text.

Line 71 The phrase 'with this activity a result of the polysaccharides [16]' makes no sense, please rewrite.

Line 72-75 This idea is confusing because of the pronouns 'they / them'. In the same phrase they means cyanobacteria and microalgae and polysaccharides. Please rewrite using shorter phrases.

Line 75 The most popular species (delete: and genera) of microalgae and cyanobacteria include..

Line 78 Another highly interesting aspect of the use of microalgae and cyanobacteria is the production of lipids, such us triacylglycerol (TAG), of special health interest for its xxxx activity (reference).

Line 110 Should Table 1 and last paragraph of section 2 be in section 3, entitled use in cosmetics? Please justify or change. 

What criteria was used due to order the information in the table? It should have a pattern, either alphabetic, by activity, by date...

Sometimes' spp' is used, sometimes sp., sometimes nothing is added; please standardize. It shoud not be in italics.

The column 'Potential uses in cosmetics' includes information about bioactives activities and formulas in which they can be added as ingredients. It would be interesting to change the title, ie. 'Potential activities and uses in cosmetics'. Include activities of all bioactives. 

Line 167 How this pigment protects from UV-A radiation? should it be taken orally, applied into the skin? How does it happen, same as lycopene does? Organize and include an small introduction if some works in the same way.

Line 246 Please, split ideas.

Lines 250-256 This paragraph seems to fit better before paragraph 247-249. The latter should include an explanation why those PUFA and vitamins are interesting in thalassotheraphy.

Line 242 Change the tittle to: Table 2.  INCI names, CAS number and description of cosmetic substances and ingredients on CosIng 243 (Cosmetic Ingredient Database)

Pool  INCI Name/CAS No. in one column only. Delete CAS No. column. Only three CAS No. are shown and it does not look nice so empty. Use capital letters only when essential.

Line 263 Is there any other references about this? any other group around the world working with this? Compare results and experience. This is not a research paper, so write accordingly. If a previous research work is cited, a reference should be added. Please check if the journal allows to cite unpublished works.

Line 269. A scheme showing the system used for cultivation beside this figure would be more informative.

Line 283 Delete (Centro de Apoyo Científico y Tecnológico a la Investigación

Line 286 Something about the mud application on the body should be written in order to make sense including Figure 2. All figures included should add some information to the paper. 

Line 300 Conclusions should be rewritten in a more coherent and robust way. Something about cyanobacteria should be said.

Line 307 References format should be consistent regarding to the use or not of 'and' before the last authors, capital letters in the titles of papers, doi, space between words, 'pp' for page numbers, chapter No and use of brackets, italics for journals' names, etc. Check the journal policy.

I hope these comments will help.

Kind regards

Author Response

Dear Reviewer,

thank you very much for you advise.

We have followed all your suggestions and comments. Unfortunately, changes have been removed when Editing English was done.

The only thing we didn´t change was table 2 (capital letters, description) as it is literally how you can find it in CosIng.

We have also rewritten the conclusions. 

Yours

Lourdes Mourelle

Reviewer 2 Report

The structure of the paper should be more coherent. Sub-headings in chapters 2 and 3 would be helpful. The titles of chapter 2 and 3 should also be coordinated since they both deal with different applications of microalgae and cyanobacteria.

Current text:

2. Interest in microalgae and cyanobacteria for health reasons

3. Uses of microalgae and cyanobacteria in cosmetics

Suggested alteration:

2. Use of microalgae and cyanobacteria for health applications

3. Use of microalgae and cyanobacteria for cosmetic applications

Chapter 4 (“Experiences on the cultivation and application of marine microalgae in thalassotherapy”) appears out of context. It is not a review but instead it describes a single experiment in present and future tense. For improved coherence, this chapter could be omitted or rewritten in a review format.

Minor comments:

Lines 9-11. Ambiguous sentence, tense etc. Rephrase.

Line 211. Grammatically incorrect. Fragmentation also needed.

Ref 25 (line 363). Publication year wrongly spelled as 199 (instead of 1995).

Author Response

Dear Reviewer,

thank you very much for your advises.

We have followed all your suggestions and comments. Unfortunately, changes have been removed when Editing English was done.

We have also changed the titles of chapter 2 and 3 following your advises.

Some new references have added.

Yours

Lourdes Mourelle

Reviewer 3 Report

Comments:1.Throughout the manuscript the level of written English is poor. There are instances of badly worded/constructed sentences e.g. gramatical and contextual related typos. Please see below comments.2.L9: remove “by humans”, it is obvious.3.L12, L44: It appears that you are missing a comma after the introductory phrase Currently.4. L13: It appears that you are missing a comma or two with the interrupter given that algal biomass. Consider adding the comma(s).5. L14. L25, L46,: Your sentence contains a series of three or more words, phrases, or clauses. Consider inserting a comma to separate the elements.6. Keywords: avoid repetition, remove cosmetics or cosmeceutics.7. L37: The spelling of photosynthesisers is a non-American variant. For consistency, consider replacing it with the American English spelling.8. L39: Misspelled word: phycocianin9. L61, 66: Misspelled word: anticarcenogenic10. L153: Misspelled word: caroten11. L162, 221: The indefinite article a may not be required with the plural noun algae in this sentence. Consider removing the article, or changing the noun to singular.12. L264: It appears that you are missing a comma after the introductory clause in this sentence. Consider adding a comma.13. L264: The word specie is not in our dictionary. If you’re sure this spelling is correct, you can add it to your personal dictionary to prevent future alerts.14. L305: buts  should be but15. Please avoid reference overkill/run-on, i.e. do not use more than 1 references per sentence. If you need to use more, make sure you state the brief relevance for each reference.16. What is the source of Figure 1?17. A schematic illustration should be given to highlight the cosmeceutical applications. For example, authors can request to reuse some Figures from the literature. Please see at: Centella et al., (2017). Marine-derived bioactive compounds for value-added applications in bio-and non-bio sectors. Journal of Cleaner Production. https://doi.org/10.1016/j.jclepro.2017.05.086.18. Authors have used different reference style in the text body. In Table 1 references are given with surnames while in the rest of the text in square brackets. Please maintain the consistency as er journal guidelines.19. Conclusion section is too little, please include further analysis with special refernce to fture perspectives.20. Referencing is not always correctly placed. There are many places where authors have ignored/missed the vol. and page numbers. Authors are encouraged to add some up to date literature references as many new reports are available from the year 2017. The inclusion of following literature is recommended.Ø  Bule et al., (2018). Microalgae as a source of high-value bioactive compounds. Frontiers in bioscience (Scholar edition), 10, 197.Ø  Centella et al., (2017). Marine-derived bioactive compounds for value-added applications in bio-and non-bio sectors. Journal of Cleaner Production. https://doi.org/10.1016/j.jclepro.2017.05.086.Ø  García et al., (2017). Microalgae, old sustainable food and fashion nutraceuticals. Microbial Biotechnology. DOI: 10.1111/1751-7915.12800.Ø  Agyei et al., (2017). Protein and Peptide Biopharmaceuticals: An Overview. Protein and peptide letters, 24(2), 94-101.Ø  Bilal et al., (2017). High-value compounds from microalgae with industrial exploitability-A review. Frontiers in bioscience (Scholar edition), 9, 319.Ø  Parsaeimehr et al., (2017). A chemical approach to manipulate the algal growth, lipid content and high-value alpha-linolenic acid for biodiesel production. Algal Research, 26, 312-322.Ø  Chandra et al., (2017). Phycobiliproteins: A Novel Green Tool from Marine Origin Blue-Green Algae and Red Algae. Protein and peptide letters, 24(2), 118-125. Ø  Ruiz-Ruiz

Author Response

Dear Reviewer,

thank you very much for your advises.

We have followed all your suggestions and comments. Unfortunately, changes have been removed when Editing English was done.

Some new references have been added.

We have also rewritten the conclusions. 

Yours

Lourdes Mourelle

Round 2

Reviewer 1 Report

Estimated authors,

You have improved notably your manuscript, however, I did not hear anything about the following comments, please change or justify:

How Biopterin glucose protects from UV-A radiation? should it be taken orally, applied into the skin?

It is needed an explanation why PUFA and vitamins are interesting in thalassotherapy. what they contribute to the skin, in what mechanisms  they interfere...?

And also:

It was nice you change the suffix 'beta' by the letter β along the text, but should be lower case instead of capital.

Figure 3 should be cited when talking about "poultices or cataplasm" because it is the moment in which the information matches (text and image).

Include citation to unpublished work as detailed in https://www.mdpi.com/journal/cosmetics/instructions:

Unpublished work, submitted work, personal communication:
4. Author 1, A.B.; Author 2, C. Title of Unpublished Work. status (unpublished; manuscript in preparation).
5.Author 1, A.B.; Author 2, C. Title of Unpublished Work. Abbreviated Journal Name stage of publication (under review; accepted; in press).

Kind regards

Author Response

Dear Reviewer,

Thank you very much for your help. We answer all your requests:

How Biopterin glucose protects from UV-A radiation? should it be taken orally, applied into the skin?

The information is included in the manuscript: It is an UVA_absorbing chromophore

It is needed an explanation why PUFA and vitamins are interesting in thalassotherapy. what they contribute to the skin, in what mechanisms  they interfere...?

We have added some information in the manuscript. Both, PUFAs and vitamins are emollients, and skin protective agents. The use of vitamins as nutritional supplements is very well-known. Special diets are very popular in thalasso centers, combining natural products, seaweed, plants, etc, (rich in vitamins and other nutrients), and nutritional supplements.

It was nice you change the suffix 'beta' by the letter β along the text, but should be lower case instead of capital.

I am very sorry, but I don´t know how to change the β to a lower case as it is the font style type from MDPI (MDPI_3.1_text.)

Figure 3 should be cited when talking about "poultices or cataplasm" because it is the moment in which the information matches (text and image).

Done

Include citation to unpublished work as detailed in https://www.mdpi.com/journal/cosmetics/instructions:

Unpublished work, submitted work, personal communication:
4. Author 1, A.B.; Author 2, C. Title of Unpublished Work. status (unpublished; manuscript in preparation).
5.Author 1, A.B.; Author 2, C. Title of Unpublished Work. Abbreviated Journal Name stage of publication (under review; accepted; in press).

We couldn’t get any unpublished work.

Thank you very much.

Kind regards

Reviewer 3 Report

The revised version reads well. I am satisfied with the revisions made. 

Author Response

Dear Reviewer,

thank you very much for your help and comments.

Kind regards